# Structure and Antiparasitic Activity Relationship of Alkylphosphocholine Analogues against *Leishmania donovani*

**DOI:** 10.3390/microorganisms8081117

**Published:** 2020-07-24

**Authors:** Humera Ahmed, Katharine C. Carter, Roderick A.M. Williams

**Affiliations:** 1Institute of Biomedical and Environmental Health Research, University of the West of Scotland School of Science and Sport High Street Paisley, Scotland PA1 2BE, UK; humera.ahmed@uws.ac.uk; 2Strathclyde Institute of Pharmacy & Biomedical Sciences, University of Strathclyde 121 Cathedral Street Glasgow, Scotland G4 ONR, UK; k.carter@strath.ac.uk

**Keywords:** *Leishmania donovani*, miltefosine, SAR

## Abstract

Miltefosine (Milt) is the only oral treatment for visceral leishmaniasis (VL) but its use is associated with adverse effects, e.g., teratogenicity, vomiting, diarrhoea. Understanding how its chemical structure induces cytotoxicity, whilst not compromising its anti-parasitic efficacy, could identify more effective compounds. Therefore, we systemically modified the compound’s head, tail and linker tested the in vitro activity of three alkylphosphocholines (APC) series against *Leishmania donovani* strains with different sensitivities to antimony. The analogue, APC12, with an alkyl carbon chain of 12 atoms, was also tested for anti-leishmanial in vivo activity in a murine VL model. All APCs produced had anti-leishmanial activity in the micromolar range (IC_50_ and IC_90_, 0.46– > 82.21 µM and 4.14–739.89 µM; 0.01– > 8.02 µM and 0.09–72.18 µM, respectively, against promastigotes and intracellular amastigotes). The analogue, APC12 was the most active, was 4–10 fold more effective than the parent Milt molecule (APC16), irrespective of the strain’s sensitivity to antimony. Intravenous administration of 40 mg/kg APC12 to *L. donovani* infected BALB/c mice reduced liver and spleen parasite burdens by 60 ± 11% and 60 ± 19%, respectively, while oral administration reduced parasite load in the bone marrow by 54 ± 34%. These studies confirm that it is possible to alter the Milt structure and produce more active anti-leishmanial compounds.

## 1. Introduction

Visceral leishmaniasis (VL) is a devastating disease for millions of people, primarily in East Africa, South Asia, South America, and Mediterranean Region. The World Health Organisation estimates that there are 200,000–400,000 people at risk of infection and that the disease is responsible for approximately 20,000–40,000 deaths/year. Despite a campaign to eliminate the disease by 2020, there are still VL cases in endemic countries such as Brazil [1]. Currently, there is no clinical vaccine for the disease, therefore control is reliant on preventive measures that prevent people from getting infected, vector control and the treatment of active cases. There are a limited number of drugs that can be used for the treatment of VL and some have been discontinued due to a reduction in clinical efficacy caused by non-compliance with treatment regimens and the emergence of drug-resistant parasites in endemic areas [2,3,4]. Ideally, new drugs to treat leishmaniasis are required but this approach is limited by the fact that leishmaniasis is a neglected tropical disease, and will not make a good investment return for pharmaceutical companies [5]. Different approaches can be used to identify new drugs, for example, repurposing clinically approved drugs for a different condition, identifying new druggable targets by having a better understanding of the molecular targets of existing antileishmanial drugs, or by designing a better therapeutic based on structure activity studies using existing antileishmanial drugs. The only oral drug for VL is the alkylphosphocholine (APC), miltefosine (Milt), a repurposed anti-cancer drug, which is the recommended second-line treatment for VL in India. Although Milt is a highly effective drug, its use is associated with a high relapse rate due to factors such as non-compliance by patients to the recommended treatment regimen, the drug’s long half-live which helps select for drug-resistant parasites [6], and naturally occurring Milt-resistant parasites [7]. For example, VL monotherapy with Milt was associated with a 20% relapse rate in Nepal at 12 months post-treatment [8]. Despite this drawback, Milt may have a role in combating the emergence of drug resistance as part of a combination treatment. It has recently been used in combination with paromomycin (PMM) or amphotericin B (AMB) to determine which regimen was most effective in curing VL and preventing relapse [9]. At 12 months post-treatment, the cure rate for treatment with AMB alone was 93.7%, AMB/Milt was 91.5% and Milt/PMM 98.6%. It is possible that changing the structure of Milt parent compound may allow the development of a drug that is still as active, has a shorter half live and is less toxic. Studies have shown that APCs with short alkyl carbon chain do have shorter half-life [10], and alterations to the hydrophilic choline head, the alkyl carbon tail and linker of this molecule have produced compounds with differential efficacy and pharmacokinetics (PK) [11,12,13].

Therefore, in this study we produced APCs modified at the head, tail and linker and tested their cytotoxicity against extracellular *L. donovani* promastigotes and intracellular *L. donovani* amastigotes using strains typed as antimony (Sb)-resistant (Sb-R) or Sb-sensitive (Sb-S) individually or in combination. The most effective compound was then used in in vivo studies to determine its ability to clear spleen, liver and bone marrow *L. donovani* parasites. Our results confirmed that the anti-leishmanial activity of APCs was dependent on their structure and that synergism was exhibited by some combinations.

## 2. Methods and Reagents

Giemsa stain was purchased from Sigma-Aldrich (Gillingham, Dorset, UK). Penicillin/streptomycin, glutamine, medium 199, DMEM, RPMI-1640, PBS pH 7.4, penicillin/streptomycin, glycine and foetal calf serum were obtained from Invitrogen, Paisley, UK. All other reagents including the APCs were analytical grade and obtained from Anatrace, via their distributor Generon Ltd., Slough, Berkshire, UK and gifted by Dr Mohamed Yaseen, UWS, UK. CMCs values were obtained from the Anatrace product description sheet for respective APCs.

### 2.1. Animals and Parasites

Age-matched inbred BALB/c female mice (20–25 g) were used in studies at Strathclyde University. Animal studies were carried out with local ethical approval and had UK Home Office approval (project license PPL60/4334).

*L. donovani* cloned strains with different Sb susceptibility backgrounds were derived from isolates obtained from VL patients at the B.P. Koirala Institute of Health Sciences, Dharan, Nepal: MHOM/NP/02/BPK282/0cl4 (Sb sensitive, Sb-S). MHOM/NP/02/BPK087/0cl11 (Sb intermediate, Sb-I) and MHOM/NP/02/BPK275/0cl18 (Sb resistance, Sb-R [13]). In addition, Milt-resistant parasites raised against these three Sb sensitivity backgrounds [14] and the Sb-sensitive strain (MHOM/ET/67:LV82) was also used in studies. Luciferase expressing promastigotes for the LV82 and Nepalese strains were prepared using previously published methods and the integrative construct (a gift from Dr D.F. Smith), designated pGL1313, contained pSSU-int fragments to facilitate integration into the ribosomal RNA locus of *Leishmania* [15].

*Leishmania donovani* promastigotes of LV82 (MHOM/ET/67:LV82), the Nepalese clinical *Leishmania donovani* isolates were cultured in complete minimum media, RPMI 1640 or HOMEM (supplemented with 20% (*v*/*v*) heat-inactivated foetal calf serum, 1% (*v*/*v*) penicillin/streptomycin, 100 µg/mL and 1% (*v*/*v*) l-glutamine) at 25 °C. The transgenic line cultures were further supplemented with Hygromycin B. Parasites were passaged weekly at a cell density of 2 × 10^5^ cell/mL in a 10 mL volume of the appropriate medium in a 25 cm^3^ tissue culture flask.

Amastigotes were cultured within bone-marrow-derived macrophages from BALB/c in Dulbecco’s modified eagle medium (DMEM supplemented with 20% (*v*/*v*) FCS, 1% (*v*/*v*) pen/strep (100 µg/mL), 30% (*v*/*v*) l-cell supernatant and 1% (*v*/*v*) l-glutamine) at 37 °C, 5% CO₂ for 72h.

### 2.2. In Vitro Cytotoxicity against L. donovani and Uninfected Macrophages

The anti-leishmanial activity of APC against *L. donovani* luciferase-expressing promastigotes was determined by adding of the appropriate 100 µL parasite line (10^6^ cells/mL) to the wells of a 96-well plate and adding 100 µL medium alone (controls) or 100 µL of APC compound (0.01 µg/mL–6.25 µg/mL, *n* = 3/treatment). The plate was incubated for 72 h at 25  °C. In combination assays, the additional compound/compounds, at a final concentration of 0.195 or 0.39 µg/mL, was added to the initial APC being tested.

In the macrophage studies, the method described in previous studies was used as follows [15]. Briefly, bone-marrow-derived BALB/c macrophages (1–2 × 10^5^) in 100 µL complete RPMI 1640 medium were added to the wells of a 96 well plate and left to adhere for 24 h at 37 °C, 5% CO₂/95% air. The medium was removed and 100 µL of the appropriate *L. donovani* luciferase-expressing promastigotes line (parasite: host cell ratio 1:10, 1:20 or 1:40) was added to each well, the ratio used reflected the infectivity of the strain and ensured that control data gave a bioluminescent signal within the same range. We used different parasite: host cell ratios in our studies to ensure that we had comparable infection rates in screening studies, an approach we have used in previous studies to improve infectivity. The variability in infectivity complicates strain to strain comparisons, but we decided to focus on achieving good infection levels in controls so that the effect of drug treatment could be determined. It is possible that intracellular macrophage parasites could have been at different developmental stages at 24 h post-infection. However, microscopical examination of parasites within host cells at 24 h post-infection, using the method described by Carter et al., 2001 [16], showed that the parasites were amastigote-like. Therefore, we assumed that the effect of compounds on the amastigote stage rather than the promastigotes stage was determined in in vitro macrophage studies.

The plate was incubated for a further 24 h, the medium removed, and each well washed with PBS to eliminate non-internalized parasites, which were monitored microscopically. When the wells were devoid of external parasites, 100 µL of the medium alone (control) or appropriate APC compound was then added (0.01–0.197 µg/mL, *n* = 3/treatment). In combination assays, the additional compound/compounds, at a final concentration of 0.024 or 0.048 µg/mL, was added to the initial APC being tested. All plates were incubated for 72 h at 37 °C, 5% CO₂/95% air. Wells with macrophage and with no drug or promastigotes added were used in control experiments.

Luciferin solution (1 µg/mL in 20 µL medium without FCS) was added to the appropriate wells of the 96-well plate at the end of the experiment and the amount of light emitted/well was measured using a luminometer (Biotek Synergy HT, relative light units) using a wavelength/bandwidth of 440/40 nm, or IVIS^®^ imaging (Spectrum Living Image system^®^, total flux, photons/sec).

The effect of drug treatment on parasite survival was determined by calculating the mean suppression in the light emitted from the drug-treated experimental sample compared to the mean control value and used to calculate the IC_50_ using Grafit^®^ software (version 5.0, Erithacus Software, East Grinstead, West Sussex, UK). The effect of drug alone on the viability of uninfected macrophages was determined using the same experimental protocol above, but cell viability was determined using an alamar blue colorimetric assay [17]. At the end of the incubation period, cells’ 10 µL resazurin solution (0.02% *w*/*v*) was added to control and drug treated cells and the absorbance of samples was read at 575 and 595 nm. The effect of drug treatment on cell survival, which correlates with the magnitude of dye reduction, was expressed as percentage viability (or alamar blue reduction [18]) and determined using the formula provided in the manufacturer’s protocol
$$\%\;alamar\;blue\;reduction=\frac{\left(\varepsilon_{\mathrm{ox}}\lambda \right)\left({A\lambda }_{1}\right)-\left(\varepsilon_{\mathrm{ox}}\lambda_{1}\right)\left({A\lambda }_{2}\right)}{\left(\varepsilon_{red}\lambda_{1}\right)\left(A^{\prime }\lambda_{2}\right)-\left(\varepsilon_{red}\lambda_{2}\right)\left(A^{\prime }\lambda_{1}\right)}x100$$

In the formula, ελ_1_ and ελ_2_ are constants representing the molar extinction coefficient of alamar blue at 575 and 595 nm, respectively, in the oxidized (ε_ox_) and reduced (ε_red_) forms. Aλ_1,_ Aλ_2_ and A’λ_1,_ A’λ_2_ represent absorbance of test and negative control wells at 575 and 595 nm, respectively. The values of % alamar blue reduction (or % viability) were corrected for background values of negative controls containing medium without cells. Percent viability values were used to calculate the IC_50_ using Grafit^®^ software (version 5.0).

The selectivity indices (SI) were determined as
$$\begin{array}{c} \mathrm{Selectivity}\;\mathrm{indices}\;\left(\mathrm{SI}\right)\\ =\frac{\mathrm{cellular}\;\mathrm{toxicity}\;\mathrm{for}\;\mathrm{uninfected}\;\mathrm{macrophage}\;i.e\;{\mathrm{mean}\;\mathrm{CC}}_{50}\;\mathrm{value}}{\mathrm{antiparasite}\;\mathrm{activity}\;\mathrm{against}\;\mathrm{promastigotes}/\mathrm{intracellular}\;\mathrm{amastigotes}\;i.e.,\;{\mathrm{mean}\;\mathrm{IC}}_{50}\;\mathrm{value}} \end{array}$$


Cross resistance (CRI_50_) indexes were determined as
$$\begin{array}{c} \mathrm{Cross}\;\mathrm{selectivity}\;\mathrm{indices}\;\left(\mathrm{CRI}\right)\\ =\frac{{\mathrm{mean}\;\mathrm{IC}}_{50}\;\mathrm{value}\;\mathrm{for}\;\mathrm{the}\;\mathrm{relevant}\;\mathrm{drug}-\mathrm{resistant}\;\mathrm{strain}\;i.e.,\;\mathrm{Sb}-I\;\mathrm{or}\;\mathrm{Sb}-R\;}{{\;\mathrm{mean}\;\mathrm{IC}}_{50}\;\mathrm{value}\;\mathrm{drug}-\mathrm{sensitive}\;\mathrm{strain}\;\mathrm{Sb}-S\;} \end{array}$$

Drug interaction profiles were determined using combenefit software^®^ [19].

### 2.3. In Vivo Cytotoxicity against L. donovani

BALB/c mice were infected by intravenous injection (tail vein, no anesthetic) with 1–2 × 10^7^
*L. donovani* strain LV82 amastigote parasites, obtained from the spleen of an infected hamster. Mice (*n* = 4/treatment) were treated with PBS pH 7.4 (controls) or APC12 (40 or 80 mg/kg) by intravenous injection (tail vein, no anesthetic, 0.2 mL) on days 7 and 8 post-infection or a single oral dose on day 7 post-infection. Milt was not given orally at a dose of 80 mg/kg because of its potential toxicity. On day 14, parasite burdens in the liver, spleen and bone marrow were determined [15]. Results are from duplicate experiments, however, if a treatment had no significant effect then it was tested in subsequent experiments.

### 2.4. Statistical Analysis

Descriptive statistics of mean and standard deviation values were used to represent data for at least three independent experiments each done in triplicate. To explore differences between control and cytotoxicity assays, T-test statistics were calculated with a statistical threshold of significance set at *p*  <  0.01 or *p* <  0.05.

## 3. Result

### 3.1. The Effect of Varied Alkyl Carbon Chain Lengths of APCs against Leishmania donovani

Nine APC series with physical modifications at the head, tail and linker (Figure 1) were screened for their antiparasitic activity against *L*. *donovani* (MHOM/ET/67:LV82). The IC_50_ and IC_90_ values showed that promastigotes were significantly more resistant to the Milt (APC16; mean IC_50_ ± SD, promastigote stage, 0.70 ± 0.00 µg/mL, amastigote stage 0.10 ± 0.01 µg/mL; mean IC_90,_ promastigote stage, 6.3 ± 0.10 µg/mL, amastigote stage, 0.90 ± 0.016 µg/mL). The activity of APC16 against uninfected macrophages was much higher and based on the IC_50_ values gave a selectivity index of 49.88 for the intracellular amastigote stage, respectively (Table 1 and Appendix A).

Decreasing the alkyl carbon chains of this molecule by two carbons produced APC14 and APC12, and increased the biological activity of the resulting compound significantly along the series when compared to promastigotes and amastigotes treated with APC16 and peaked with APC12 (IC_50_ and IC_90_ of 0.163 ± 0.00 µg/mL, 0.009 ± 0.00 µg/mL, and 1.47 ± 0.00 µg/mL, and 0.081 ± 0.01 µg/mL for promastigotes and intracellular amastigotes, respectively; (*p* < 0.01 and *p* < 0.01; Table 1). The SI for APC12 was and 3870.77. Molar concentrations of all compounds are detailed in Table 1.

Decreased alkyl APC carbon chain length was synonymous with decreased hydrophobicity and increased readiness to form micellar structures that form pores on cell membranes to produce death by leakage [20].

Increasing hydrophobicity of APC12, by introducing a double bond between the first and second carbon atom on the APC12 tail (APC11UPC; Figure 1), allowed the formation of lamellar structures and not micelles [20], significantly reduced efficacy relative to APC12 for promastigotes and not amastigotes (*p* < 0.01; Table 1 and Appendix A). In contrast, the addition of two alkyl carbon chains to the choline backbone of APC12, each with six alkyl carbons atoms, to reduce their ability to form micelles and increased hydrophobicity (APC11PC; Figure 1 [20]) significantly reduced efficacy relative to APC12 for promastigotes and amastigotes, respectively (*p* < 0.01 and *p* < 0.01, respectively; Table 1 and Appendix A). Intriguingly, the death of *L. donovani* promastigotes and intracellular amastigotes judged by their IC_50_s, occurred below, near and above the threshold concentrations micelles are formed, also called the critical micellar concentration (CMC) for APC12, APC14 and APC16, respectively (Table 1 and Appendix A). These results suggested that APCs with reduced hydrophobicity or increased ability to form micelles were effective anti-leishmanials.

Next, the antiparasitic activity of APC12, APC14 and APC16 against three Nepalese *L. donovani* clinical isolates with different inherent susceptibilities to antimony, reflective of strains present in endemic communities in the ISC, was tested. We found that the Sb-S, Sb-I and Sb-R [14] strains were also killed by the APC analogues and that efficacy was influenced by alkyl carbon chain length. APC16 and APC12 were the least and most toxic with death occurring below CMC for APC12- and APC14-treated Sb-S and Sb-I promastigotes, and above CMC for APC16-treated Sb-R promastigotes and amastigotes (Table 2 and Appendix A). Cross-resistance indexes (CRI_50_) for Sb-resistant promastigotes and amastigotes to APCs ranged from 0.48–1.77 and 0.77–2.33, respectively, showing that the CRI_50_ value reflected the antimony resistance of the strain (Table 2; Appendix A). Further, the selectively index (SI_50_) showed that APC12 and APC14 were more effective than APC16 and were safe compounds for in vivo experiments in *Leishmania* mice models (Table 2).

Next, the antiparasitic activity of APC12, APC14 and APC16 against Milt-resistant parasites induced against the three Nepalese *L. donovani* clinical isolates with different inherent susceptibilities to antimony was tested. The efficacy of the APC against parasites resistant to Milt-resistant amastigotes and not promastigotes was influenced by alkyl carbon chain length, with APC12 being the most toxic (Table 3). The cross-resistance index (CRI_50_) for amastigotes and not promastigotes to APCs reflected the antimony resistance of the strain (Table 3; Appendix A). Further, the selectively index (SI_50_) showed that APC12 was more effective than against the Sb-sensitivity lines and less so against the Sb-resistant parasites. This result suggests that APC12 was a safe compound for in vivo experiments in *Leishmania* mice models (Table 2).

All of the compounds were more active against the LV82 *L*. *donovani* laboratory strain compared to the Nepalese clinical isolates (compare Table 2 and Table 1), which may reflect the temporal difference when these strains were isolated.

### 3.2. The Effect of Charge on Tailed Molecules against Leishmania donovani

Modifications to the APC12 head, by removing the P-atom, to produce a cationic amphiphile, where the N-atom attached to one or two 12 alkyl carbons to produce DA and DAB, respectively (Figure 1), resulted in similar activity to APC12 based on IC_50_ values for DAB against the amastigote and promastigote stages (Table 1). The anionic amphiphile, formed by the removal of the N-atom from APC12 to produce PO (Figure 1), was inactive against *L. donovani* compared to controls (Table 1).

### 3.3. The Effect of Charge Separation on Tailed Molecules against L. donovani

Increasing the number of alkyl carbons in the linker between the N- and P-atoms in APC12 from two to six (APC12P6C; Figure 1) significantly decreased the compound’s anti-leishmanial activity against both promastigotes and amastigotes relative to APC12 (*p* < 0.01; Table 1 and Appendix A).

### 3.4. Studies to Determine if APC can Act Synergistically against L. donovani

The combination of drugs with multiple targets can produce more effective antimicrobials and the discovery that APCs induced death below, near or above CMC against *L. donovani* allowed us to investigate their interaction as mixed APCs, namely, APC12 with APC14 or APC16, by applying the Loewe additivity model to viability data using Combenefit [19]. APC12 mixed with APC14 interacted in a predominantly synergistic manner against promastigotes (Figure 2A; Appendix A), but was antagonistic against amastigotes intracellularly within macrophage (Figure 2C, Appendix A). In contrast, APC12 and APC16 had significant antagonistic interactions against both life cycle forms (Figure 2B,D; Appendix A) suggesting that identifying synergistic combinations was unpredictable and unrelated to the physical properties of APCs, but required systematic screening of all possible drug combination ratios within the physiological level for the model organism.

### 3.5. In Vivo Efficacy of APC12 against L. donovani

In this study, we decided to assess the in vivo anti-leishmanial activity of the most active analogue, APC12, in a murine model of VL. A dose at 40 mg/mL caused a significant reduction in parasite numbers in the spleen (*p* < 0.05) and liver (*p* < 0.01), but not in the bone marrow given by the intravenous route, and had no significant activity when given by the oral route (Table 4). Doubling the drug dose administered by the oral route to 80 mg/kg still did not result in a significant reduction in parasite burdens in all three sites (Table 4). In contrast, oral treatment with Milt at a dose of 40 mg/kg resulted in a significant reduction in parasite numbers in both the spleen (*p* < 0.05) and liver (*p* < 0.01, Table 4). These results probably indicate that APC12 has poor bioavailability by the oral route.

## 4. Discussion

In this study we have shown that the activity of APCs against *L. donovani* promastigotes was dependent on its alkyl carbon chain length, with a 12-alkyl carbon analogue that is zwitterionic or cationic (DA) charge on the head being most effective. The candidate APC12 was 10-fold and 4-fold more active than the Milt (APC16) against promastigotes and intracellular amastigotes, respectively.

The significant activity of APC12 against Sb- and Milt- resistant parasites, suggests that it can be used for treatment of leishmaniasis caused by these parasites in clinics. This observed phenomenon from the APC series in this study is pathogen-specific, as a parallel study using the same compounds against the parasitic protist, *Acanthamoeba*, showed that APC16 was the most active [21]. Studies from elsewhere have also produced mixed results. For example, short- and long-chain APCs were most effective in reducing tumour development and against fungal infections, respectively [21,22,23,24,25]. The reason for this is speculative and possibly depends on the pathogen’s membrane infrastructure, where membranes with fatty acids with long chains are easily disrupted with complementary APCs. For example, the predominant fatty acids present in membrane phospholipids of *Acanthamoeba* trophozoites and *Leishmania* promastigotes are ~28–30 carbons and 30–40 carbon atoms, respectively, [26,27], and these are most sensitive to APC16 [22]. Nevertheless, the link between the number of the alkyl carbon chains in APCs and their CMCs and hydrophobicity is well-established [27,28,29]. Decreasing alkyl carbon chain length (APC16-APC12) results in alterations in the physical properties, such as hydrophobicity and CMC [26]. Introducing unsaturation (APC11UPC) produced the expected results except for widening the charge distance from 33 to 70Å to produce the flexible molecule, APC126PC. The reduced CMC for APC126PC was accompanied by a higher biological activity, possibly because the flexibility produced at the head affected its orientation and packing density on the biological membranes [30,31]. There are specific but limited studies on this theme to support this premise. For example, erucylphosphocholine [31], edelfosine [31], ilmofosine [32] and perifosine [28,33,34,35,36], with 21, 18, 16, 18 alkyl carbon atoms tails modified with a cis double bond, oxygen, sulphur or a piperidine ring at the head, respectively, have produced more effective antileishmanial activity against *Leishmania* spp. compared to Milt. These differences are perhaps dictated by their ability to form micelles at a lower critical micellar concentration (CMC; [30,31,37]. Other death mechanisms of APCs include their ability to rapidly reversal the net negative charge of the plasma membrane to produce shock [38,39]. Thus, we can expect that molecules with similar charge as the membrane have limited toxicity. This is confirmed by the limited activity of PO in this study. Consistent with this is, that molecules that reduce the net negative charge of the membrane potential of *Leishmania* spp are normally effective anti-leishmanials [40,41]. This suggests that the net positive charge of the anionic molecules, DA and DAB, had reversed the net charge of the parasite membrane to produce death comparable to APC12. One setback is that the low selective index suggests toxicity issues if used for in vivo studies. Nevertheless, the molecules used in this study produced death below (APC12), near (APC14) and above (APC16) the CMC. In bacteria, this produced three different death mechanisms [42], but evidence for this in *Leishmania* is limited. However, if this were the case for *Leishmania*, we postulate that mixtures of APCs with different and dissimilar CMC-APC relationships should be synergistic.

Combination treatment with APC12 and APC14 or APC16 was only synergistic for promastigotes and not amastigotes, which could reflect differences in drug uptake by macrophages or the ability to reach the intracellular parasites within the host cell. Generally, APCs mixtures can produce unique molecular species with physical characteristics such as surface tension, osmotic pressure, solubility and ease to form micelles which are significantly different from their individual constituents [43,44,45]. It shows that studies should focus on the intracellular amastigote stage, as this is what is clinically relevant and that we need to understand how drug combination functions to produce better combination treatments. These types of regimens are very important as liposomal amphoptericin B and Milt combination treatment acted synergistically, giving a cure rate of >95% [46,47,48,49].

Further, our in vitro screening studies identified APC12 as the most potent compound, but this high activity was not reflected in a murine VL model when administered orally. This suggests that the intestinal epithelium may be a significant barrier for oral absorption of APC12. Generally, the alkyl chain and zwitterionic head group in hydrophilic APCs, e.g., APC12 and APC16 modulates the activities of transporters, e.g., P-glycoprotein, the human intestinal peptide transporter (PepT-1) and the monocarboxylic acid transporter (MCT-1) and the intercellular spaces between tight junctions (paracellular transport in membranes to navigate through the lipid bilayer of Caco-2 cell monolayer to increase bioavailability [50,51,52,53]. Based on this information, the reason for the intestine of mice being a formidable barrier against the delivery of APC12 into systemic circulation is not readily apparent. Nevertheless, the drug bioavailability of drugs is influenced by the characteristics of the site where the compound is absorbed and for the gut, factors such as (a) the absorbing surface area, bacterial flora, motility, pH, mucus thickness and food intake [54], (b) drug concentration [52] and (c) structure [55] are compounding factors. For example, Milt can be non-saturable and saturable in a two-component population PK model below and above a threshold concentration of 50 μM (20.4 μg/mL) [50,51,52,53], while the ability of a series of 2-alkoxy-3-alkylamidopropylphosphocholine derivatives to alter cell membrane fluidity was a function of the lengths of their alkyl carbon chain lengths and not their ability to form micelles [55]. Studies to compare the PK profile of APC12 and Milt would show whether PK differences are responsible for the low oral activity of APC12. Nevertheless, our study and previous studies show that oral treatment Milt at 40 mg/kg (this study) or 25 mg/kg on days 7–11 [56] produced a significant reduction in *L. donovani* parasite burdens for WT (this study; *p* < 0.0010) and an Sb-sensitive and a Sb-resistant strain, [56].

In conclusion, we postulate that further studies are still required to investigate the non-translational nature of our in vitro cytotoxicity assays in our in vivo mice model.

## Figures and Tables

**Figure 1 microorganisms-08-01117-f001:**
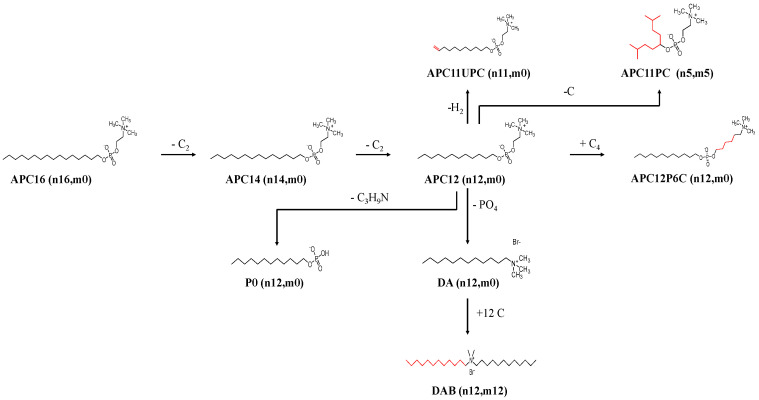
Diagram to show the changes made to miltefosine (Milt) to produce three alkylphosphocholine (APC) series. The one tailed hexadecylphosphocholine, with sixteen alkyl carbon atoms on the tail (n-16, m-0, designated APC16) was progressively reduced by the removal of two alkyl carbons to give tetradecylphosphocholine, APC14, and dodecylphosphocholine, APC12. To APC12, a cis double bond was added between the first and second atoms, 10-Undecylenyl-1-phosphocholine, (APC11UPC) or its 12 alkyl carbon atoms reduced to two five alkyl carbon chains (n-5, m-5) attached to a carbon on the phosphoryl group to produce 2, 8-Dimethyl-5-Nonylphosphocholine (APC11PC). The positive and negative charge on the N- and P- atoms, respectively, were separated using four alkyl carbon atoms to give APC12P6C, or the N-atom (trimethyl amine moiety) or P-atom was removed to give dodecylamine, docecyltrimethyl ammonium bromide (DA) and mono –n-dodecyl phosphate (PO), respectively. Twelve alkyl carbons were added to produce didodecyldimethyl ammonium bromide (DAB). Significant structural changes are shown in red.

**Figure 2 microorganisms-08-01117-f002:**
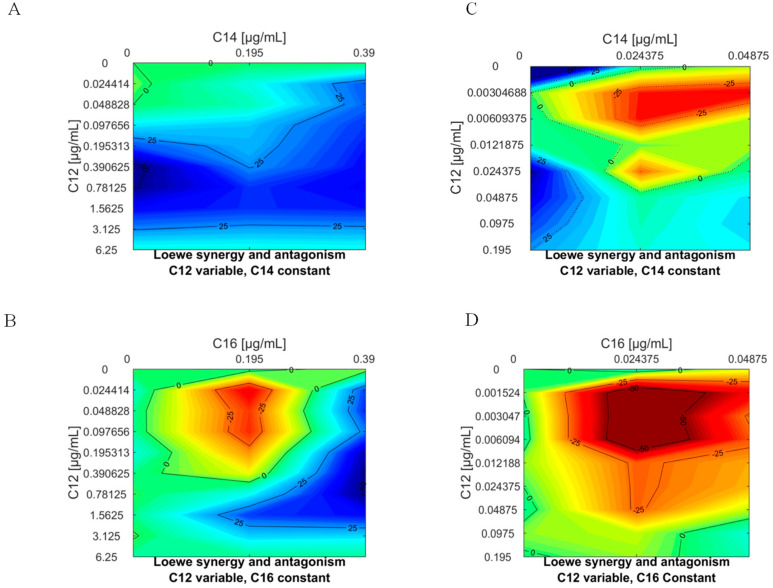
Contour map of the interaction based on antileishmanial activity between APCs mapped-out with the Loewe model. The Combenefit software produced an interaction profile for APC12 (0.02–6.25 µg/mL for promastigotes (**A**,**B**) and 0.001–0.19 µg/mL for amastigotes (**C**,**D**) mixed with APC14 or APC16 at 0.195 or 390 µg/mL for promastigotes and 0.02 or 0.04 µg/mL for amastigotes. Synergistic (blue), additive (green to yellow) and antagonism (red) interaction were noted.

**Table 1 microorganisms-08-01117-t001:** The activity of APCs against *L. donovani* parasites with different inherent susceptibilities to antimony, i.e., Sb-sensitive, Sb-intermediate or Sb-resistant, at the promastigotes and intracellular amastigotes stage. The activity of APCs against bone-derived macrophages (1 × 10^5^ cells) from BALB/c mice and *L. donovani* LV82 promastigotes (10^6^ cells/mL) and intracellular amastigotes (1 × 10⁶–1 × 10⁷ cells) using bone-marrow-derived macrophages from BALB/c mice were cultured in the presence of medium alone (controls) or different concentrations of APCs for 3 days (*n* = 3/treatment) at 37 °C/5%CO_2_, 25 °C and 37 °C5%CO_2_ respectively. The effect of treatment on parasite survival was determined by determining the mean suppression in parasite growth for each experimental value compared to the mean control value. The data were then used to determine the mean IC_50_ using Grafit software, and Selectivity index (SI_50_). CMC, critical micellar concentration values were provided by Anatrace.

Compound	Chain Length and Number (n,m)	Charge	Promastigotes µg/mL (µM)	Amastigote µg/mL (µM)	Macrophage µg/mL (µM)	Selectivity Index (SI_50_)	Promast: Amast Ratio	Molecular Mass	CMC µg/mL (µM)
	**Modification of tail**								
**APC12**	n-12; m-0	Zwitterionic	** 0.16 ± 0.00 (0.46 ± 0.00)	** 0.01 ± 0.00 (0.026 ± 0.00)	35.38 ± 1.74 (100.64 ± 4.95)	3870.77	16.46	351.5	351.50 (1.00)
**APC14**	n-14; m-0	Zwitterionic	** 0.20 ± 0.00 (0.53 ± 0.00)	** 0.03 ± 0.02 (0.066 ± 0.05)	40.89 ± 4.42 (107.75 ± 11.65)	1632.58	7.97	379.5	45.54 (1.2 × 10^-5^)
**APC16**	n-12; m-0	Zwitterionic	0.70 ± 0.00 (1.72 ± 0.00)	0.10 ± 0.01 (0.25 ± 0.02)	5.08 ± 8.29 (12.47 ± 20.34)	49.88	7.00	407.5	0.41 (1.3 × 10^-5^)
**APC11PC**	n-6; m-5	Zwitterionic	^b^ >6.25 ± 0.06 (> 18.63 ± 0.18)	^b^ 0.19 ± 0.08 (0.55 ± 0.24)	61.90 ± 0.01 (184.56 ± 0.03)	335.56	>33.60	335.4	NA
**APC11UPC**	n-12; m-0	Zwitterionic	^b^ 1.80 ± 0.02 (5.37 ± 0.06)	0.01 ± 0.06 (0.04 ± 0.18)	54.00 ± 0.13 (161.00 ± 0.39)	4025.00	135.34	335.4	NA
	**Modification of the linker**								
**APC12P6C**	n-12; n-0	Zwitterionic	^b^ 33.50 ± 0.06 (82.21 ± 0.15)	^b^ 1.01 ± 0.07 (2.47 ± 0.17)	14.10 ± 1.08 (34.60 ± 2.65)	14.01	33.33	407.5	NA
	**Modification of the head**								
**DA**	n-12; m-0	Cationic	^b^ 0.09 ± 0.00 (0.05 ± 0.00)	0.47 ± 0.00 (2.54 ± 0.00)	37.60 ± 0.01 (202.91 ± 0.05)	79.88	0.18	185.3	NA
**DAB**	n-12; m-12	Cationic	^b^ 0.04 ± 0.05 (0.091 ± 0.11)	^b^ 0.06 ± 0.00 (0.119 ± 0.00)	82.76 ± 0.01 (178.91 ± 0.02)	1503.45	0.79	462.6	NA
**PO**	n-12; m-0	Anionic	^b^ >6.25 ± 0.01 (> 25.07 ± 0.04)	^b^ >2.00 ± 0.01 (> 8.02 ± 0.04)	100.00 ± 0.01 (401.11 ± 0.04)	>50.00	>3.12	249.31	NA

* *p* < 0.05, ** *p* < 0.01 compared to APC16. ^a^
*p* < 0.05, ^b^
*p* < 0.01 compared to APC12, respectively. NA data not available.

**Table 2 microorganisms-08-01117-t002:** The activity of *L. donovani* parasites with different inherent Sb susceptibilities which had induced resistance to Milt, i.e., Milt-Sb-sensitive, Milt-Sb-intermediate or Milt-Sb-resistant, at the promastigotes and intracellular amastigotes stage. The activity of APCs against bone-derived macrophages or *L. donovani* parasites with different inherent susceptibilities to antimony, i.e., Sb-sensitive, Sb-intermediate or Sb-resistant, at the promastigotes and intracellular amastigotes stage was determined. Bone-marrow-derived macrophages from BALB/c mice (x/mL), *L. donovani* promastigotes (x/mL) or macrophages infected with *L. donovani* promastigotes for 24 h were cultured with medium alone (controls) or different concentrations of APCs for 3 days (*n* = 3/treatment). The cytotoxicity of the APCs against uninfected macrophages was determined using an alamar blue colorimetric assay assay whereas a luciferase assay was used for parasite studies. The suppression in uninfected macrophage or parasite survival was determined by comparing individual experimental values with the relevant mean control data. These values were used to determine the mean IC_50_, using the Grafit software, Cross Resistance Index (CRI_50_) and Selectivity index (SI_50_) shown.

Strain	Antimony Resistance	Drug	Mean IC_50_ µg/mL (µM)		Promast: Amast Ratio	Cross-Resistance Index (CRI_50_) (Promast/Amast)	Selectivity Index (SI_50_)
			Promastigotes	Amastigotes			
**282/4**	**Sb-sensitive**	APC12	** 0.55 ± 0.02 (1.56 ± 0.06)	** 0.26 ± 0.01 (0.74 ± 0.03)	2.09	NA/NA	136.08
		APC14	** 0.44 ± 0.07 (1.16 ± 0.18)	** 0.12 ± 0.02 (0.32 ± 0.05)	3.64	NA/NA	340.75
		APC16	0.84 ± 0.05 (2.06 ± 0.12)	0.31 ± 0.02 (0.76 ± 0.05)	2.68	NA/NA	16.39
**087/11**	**Sb-intermediate**	APC12	** 0.42 ± 0.02 (1.19 ± 0.06)	** 0.20 ± 0.07 (0.57 ± 0.20)	2.11	0.76/0.77	176.90
		APC14	** 0.21 ± 0.00 (0.55 ± 0.00)	** 0.19 ± 0.01 (0.50 ± 0.03)	1.05	0.48/1.58	215.21
		APC16	1.49 ± 0.6 (3.66 ± 1.47)	0.29 ± 0.01 (0.71 ± 0.02)	5.15	1.77/1.76	17.52
**275/18**	**Sb-resistant**	APC12	** 0.49 ± 0.03 (1.39 ± 0.09)	** 0.23 ± 0.01 (0.65 ± 0.03)	2.10	0.89/0.88	153.83
		APC14	** 0.37 ± 0.05 (0.97 ± 0.13)	** 0.28 ± 0.00 (0.74 ± 0.00)	1.31	0.84/2.33	146.04
		APC16	1.26 ± 0.06 (3.09 ± 0.15)	0.51 ± 0.00 (1.25 ± 0.00)	2.49	1.50/1.65	9.96

* *p* < 0.05, ** *p* < 0.01 compared to APC16. NA Data not available.

**Table 3 microorganisms-08-01117-t003:** The activity of APCs against bone-derived macrophages or *L. donovani* parasites with different inherent Sb susceptibilities which had induced resistance to Milt, i.e., Milt-Sb-sensitive, Milt-Sb-intermediate or Milt-Sb-resistant, at the promastigotes and intracellular amastigotes stage was determined. Bone-marrow-derived macrophages from BALB/c mice, *L. donovani* promastigotes (x/mL) or macrophages infected with *L. donovani* promastigotes for 24 h were cultured with medium alone (controls) or different concentrations of APCs for 3 days (*n* = 3/treatment). The cytotoxicity of the APCs against uninfected macrophages was determined using an alamar blue colorimetric assay assay, whereas a luciferase assay was used for parasite studies. The suppression in uninfected macrophage or parasite survival was determined by comparing individual experimental values with the relevant mean control data. These values were used to determine the mean IC_50_, using the Grafit software, Cross Resistance Index (CRI_50_) and Selectivity index (SI_50_) shown.

Strain	Antimony Resistance	Drug	Mean IC_50_ Value µg/mL (µM)		Promast: Amast Ratio	Cross-Resistance Index (CRI_50_) (Promast/amast)	Selectivity Index (SI_50_)
			Promastigotes	Amastigotes			
**282/4**	**Milt-Sb-sensitive**	APC12	>125.00 ± 0.00 (>355.60 ± 0.00)	**0.08 ± 0.17 (0.23 ± 0.48)	>1562.5	NA/NA	442.25
		APC14	>125.00 ± 0.00 (329.40 ± 0.00)	>1.73 ± 00 (>4.56 ± 00)	>72.25	NA/NA	>23.64
		APC16	>125.00 ± 0.00 (306.70 ± 0.00)	>1.61 ± 00 (>3.95 ± 00)	>77.64	NA/NA	>3.16
**087/11**	**Milt-Sb-intermediate**	APC12	** 184.00 ± 0.10 (0.52 ± 0.22)	** 6.17 ± 0.01 (17.55 ± 0.03)	29.82	>1.47/77.13	5.73
		APC14	232.00 ± 0.30 (0.63 ± 0.11)	10.03 ± 0.00 (26.43 ± 0.00)	23.13	>1.86/> 5.95	4.08
		APC16	242.00 ± 0.24 (0.56 ± 0.12)	12.65 ± 0.00 (31.04 ± 0.00)	19.13	>1.94/> 7.86	0.40
**275/18**	**Milt-Sb-resistant**	APC12	>125.00 ± 0.00 (>355.60 ± 0.00)	** 4.45 ± 0.01 (12.66 ± 0.03)	>28.09	1.00/5.62	7.95
		APC14	>125.00 ± 0.00 (329.40 ± 0.00)	** 34.85 ± 0.02 (91.83 ± 0.05)	3.59	1.00/> 20.144	1.17
		APC16	>125.00 ± 0.00 (306.70 ± 0.00)	>61.36 ± 0.00 (>150.58 ± 0.0)	2.04	1.00/> 38.11	>0.08

* *p* < 0.05, ** *p* < 0.01 compared to APC16; NA Data not available.

**Table 4 microorganisms-08-01117-t004:** The in vivo activity of different formulations against *L. donovani* spleen, liver and bone marrow parasite burdens. *L. donovani* infected mice (*n* = 4 or 5) were treated with medium alone (intravenous route), miltefosine (Milt, oral), or APC12 (oral or intravenous route, IV) on day 7 post-infection and parasite burdens then assessed on day 14 post-infection. The mean percentage suppression ± SD in parasite burdens is shown in parentheses. **p* < 0.05, ** *p* < 0.01 vs. control, ^a^
*p* < 0.05, ^b^
*p* < 0.01 MIL vs. APC12, ^c^
*p* < 0.05 APC12 40 vs. 80 mg/kg.

Treatment	Mean Parasite Burden ± SD
Spleen	Liver	Bone Marrow
**Experiment 1: Oral administration**
Control	198 ± 62	1000 ± 286	225 ± 100
MIL 40 mg/kg oral	73 ± 68 *(53 ± 31)	433 ± 46 **(40 ± 25)	197 ± 157(30 ± 35)
APC12 40 mg/kg oral	275 ± 96 ^b^(8 ± 18)	880 ± 276 ^a^(36 ± 34)	210 ± 90(54 ± 34)
**Experiment 2: Intravenous administration**
Control	305 ± 97	1299 ± 158	458 ± 129
APC12 40 mg/kg IV	112 ± 30 *(60 ± 11)	434 ± 197 **(60 ± 19)	555 ± 254(8 ± 14)
APC12 80 mg/kg oral	257 ± 73 ^c^(13 ± 27)	939 ± 170 ^c^(27 ± 16)	607 ± 115(2 ± 4)

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
