# Peer review of "Structure and Antiparasitic Activity Relationship of Alkylphosphocholine Analogues against Leishmania donovani"

_microorganisms, 2020, doi:10.3390/microorganisms8081117_

Round 1

Reviewer 1 Report

This manuscript tests a series of molecules related to miltefosine on mice infected with L. donovani.

I was consulted as a statistical reviewer but the manuscript offers no description of the tools used to analyze the data. In that sense, this ms is fundamentally flawed. Also there is no such thing as "statistically decreased signficance".

Overall the manuscript also needs to be carefully edited and revised to ensure all relevant details are included and that the text flows logically.

These are a few examples of systematic problems that need to be addressed if this ms is revised:

Line 30 the World HEALTH Organization

Line 34 check on preventive measures on preventing people

Line 68 How many mice?

Line 70 When was the ethical approval granted_

Author Response

We would like to thank the reviewers for the work they have completed in reviewing our manuscript of on “The Structure and Antiparasitic Activity Relationship of Alkylphosphocholine Analogues against Leishmania donovani’ .  We have addressed the individual points you have raised below.

This manuscript tests a series of molecules related to miltefosine on mice infected with L. donovani.

  1. I was consulted as a statistical reviewer but the manuscript offers no description of the tools used to analyze the data. In that sense, this ms is fundamentally flawed.

We have now added the  statistical tests we used to analyse the data have been added a new section, Section 2.4 titled  Data analysis in the Methods section in the resubmitted manuscript (Lines 146-150).

  1. Also there is no such thing as "statistically decreased significance".

We agree with the reviewer that this statement did not make sense and apologise  we have amended the foot note of Tables 1 and 2 from ‘statistically decreased significance’ to ‘statistical significance’ (Lines 200 and 225).

  1. Overall the manuscript also needs to be carefully edited and revised to ensure all relevant details are included and that the text flows logically.

We have edited the manuscript to make the text flow better we have highlighted all these changes in the amended manuscript.

These are a few examples of systematic problems that need to be addressed if this ms is revised:

We apologise for these errors and the highlighted the changes we have made to address  the structure and organization of the manuscript.

Line 30 the World HEALTH Organization

Line 30 We have changed  the text to ‘World Health Organisation’ instead of ‘World Organisation’. 

  1. Line 34 check on preventive measures on preventing people

Line 34 This error has been addressed and the sentence now reads ‘disease therefore control is reliant on preventive measures that prevent people from getting infected’ instead of ‘disease therefore control is reliant on preventive measures on preventing people from getting infected’

  1. Line 68 How many mice?

Line 68 We did not give the exact number of mice used in each treatment in this section as this detail the source and age of the animals.  The exact number of mice used in each treatment within an experiment is given in the section on ‘In vivo cytotoxicity against L. donovani (line 141).

  1. Line 70 When was the ethical approval granted_

Ethical approval for in vivo studies in the UK is given by two bodies, the local ethical review body of the particular university and UK Home Office.  The manuscript contains the number of the project licence that gave the authority for this work so it can be tracked i.e. project license PPL60/4334. Local ethical approval for the project licence application was given in March 2012 and Home Office approved of this licence was on 19th June 2012.  We have realised that experiments also used project licence PF669CAE8, this had local ethical approval in January 2017 and was Home Office approval on 254th May 2017.  We have amended the text ‘to Animal studies were carried out with local ethical approval and had UK Home Office approval (project license PPL60/4334) to ‘Animal studies were carried out with local ethical approval and had UK Home Office approval (project licenses PPL60/4334 and PF669CAE8)’ (lines 85-86). We have not added the dates as this would involve adding 4 dates.

Reviewer 2 Report

The manuscript Structure and Antiparasitic Activity Relationship of Alkylphosphocholine Analogues against Leishmania donovani describes the evaluation of the anti-leishmanial potential of Alkylphosphocholine analogues. The premise for the article is scientifically sound, the search for Leishmania-optimized miltefosine derivatives.

Overall the message is clear and the manuscript is easy to read.

The introductory section is objective and provides sufficient information for the comprehension of the scientific rational. Still some background justification for some options in the manuscript would be beneficial. Why to use antimony resistant parasites in this context? What is the expected take home message from these specific parasites in the context of miltefosine derivatives? Would not be preferable to use miltefosine resistant parasites? Considering that the molecules are miltefosine derivatives and the changes on APCs subtle, what were the expectations on the possibility of synergism?

Concerning the material and methods used I have some doubts/concerns:

In line 88 please clarify if the plates were incubated at 25ºC or 27ºC.

Concerning the macrophage infections for the purpose of evaluating the activity of the APCs against intracellular amastigotes I have some doubts. No reference to passages or days of culture are made. How many days of culture have the promastigotes that were used for the in vitro assays? How was the parasite maintenance done (starting inoculum and maintenance schedule)?

In the context of the macrophage infection, there is in line 94 there a reference to 1x106-1x107 as the number of parasites added to each well, please clarify this 10 fold difference in inoculum. If strain specific inoculums for infection ratios are used please describe the rational for choice of ratio of infection. Considering that 1-2x105 macrophages were used inoculums an infection ratio of 100 or 50 parasites per cell seems an overkill. Please justify this option. It is difficult to compare different strains/species because the infectivity of the parasites and the capacity to modulate the macrophage biology can have an impact in the sensitivity to drugs. Thus it is important to know the rational for the macrophage infections. How inter-strain infection normalization was achieved? Another aspect that I would like to see clarified was the 24 hour infection. Why 24 hour infection? Would not be better a short (4 hour) infection, than wait for 24 hours before adding the drug to enable the uptake of all the parasites and enable the differentiation. From our experience this originates much more reproducible data. As the test is described, the continuous infection for 24 hours, media removal and then treatment, there is the risk having drugs that are killing parasites that are attached to the surface or parasites that are undergoing differentiation. This is not amastigote killing and might contribute to false discovery rate because the objective of the assay is to detect compounds that act on amastigotes and not compounds that kill promastigotes or interfere in the differentiation.

Please define which was the statistic approach used from the graphpad softare package to determine the IC50.

For clarity would recommend a more appropriate description of the selectivity index (Line 120). Would suggest: “The selectivity indices (SI) were determined as the quotient between the cellular toxicity determined in the uninfected cells (CC50) and the anti-parasite activity in infected macrophages determined by the IC50.” Please confirm that no mistake exists in table 1. For example for APC12 if I use the data on the table to calculate SI I get 3870 (100,64/0,026) and not the 3535 reported.

I had some difficulty to understand the rational for the in vivo experiments. Foremost I think that it was missing a PK evaluation to adjust the treatment scheme for the drug. At the moment in our team we are not testing in vivo any compounds without PK information. This helps to adjust the treatment scheme maximizing the chance of having good data. If the goal of IV administration was to confirm in vivo anti-parasitic capacity, why wait 7 days? This will probably “dilute” the effect by the multiplication of the surviving parasites, no? What is the rational for the 7 day waiting period, I would kill the animals maximum 4 days after administration. Please describe briefly the determination of parasitaemia.

No reference is made in the section of material and methods of how critical micellar concentration was determined.

Concerning the result section:

Line 136-137, for accuracy it should read “Three the APC series with physical modifications at the head, tail and linker (Fig. 1) were screened for their antiparasitic activity against L. donovani (MHOM/ET/67:LV82) expressing luciferase”. Correct? To my understanding all the tests were performed using the luciferase transfected parasites. It is different from doing in Wild Type because the transfection process will involve a selection that might alter the basic characteristics of the parental population.

Line 143 it states that “APC14 and APC12, increased the biological activity of the resulting compound significantly” How was this significance calculated, what was the statistical test? Table 1, 2 and 3 all have statistical analysis but I could not find reference to any statistical test in methods section. Along the test significance claims are made without the appropriate support for what is being compared. For example, Line 150 “…APC12 tail (APC11UPC; Fig 1), allowed the formation of lamellar structures and not micelles [18] significantly reduced the efficacy related to APC12 (Table 1 and Fig S1)”. When I look to Table 1 the statistical analysis present is related to APC16, so not only I have no way of knowing what is being compared, (Promastigotes? Amastigotes?) looking at the amastigote data the most relevant I would say they are very similar. This lifts another question, if I consider Selectivity index APC11UPC is the best derivative, why was not considered for further studies?

As stated above I see no added value in the testing to Sb resistant strains, more so when the studies are performed in transfected parasites.

Concerning section 3.2 please try to explain in the discussion section the interesting difference between DA and DAB in amastigotes.

The synergism section is interesting showing that the synergism correlates seen for promastigotes are not always reportable for the intracellular assay suggesting as the authors state that combination studies should focus on the intracellular stage (line 268). If the N/A I table 1 means “not active” than APC11UPC would have been a very nice compound to evaluate in the context of synergism because is active but not forming micelles.

The in vivo data could be improved if PK of the APC12 was available, the fact that APC12 did not work upon oral administration is disappointing and limits the value of APC12 as oral drugs are a priority in Leishmania drug development. The use of a single administration is baffling, I have used miltefosine 20 mg/kg for 20 days without any side effect and has complete clearance of infection in all quantifiable compartments. I would recommend quantify the compound of interest in the blood after administration, perform a PK and then adjust the treatment accordingly, maybe bi-daily administration is what is required to have sufficient bio-accumulation in the blood and probably one administration is not sufficient for this to happen. A PK would greatly improve the understanding of the apparent lack of activity upon oral administration. The fact that works after IV administration is a confirmation that if the molecule is on the blood it would work so this should be taken as encouraging.

Supplemental information:

It is very nice to see all the curves. In figure S2 the curve for APC12 in graph D is not sigmoidal having a plateau at 50%. How was the EC50 determined in this situation?

Minor –

All the EC50 and EC90 in the text and tables should be normalized to the same format using either two or three decimal cases. Sometimes you have standard deviation with more decimal cases that the actual reported value. For example line 140 “6.3 ± 0.10 µg/ml” or “0.90 0.10 µg/ml”.

Table 1 – The title is not completely descriptive of the table, much more than just SI and EC50 are depicted. It is also missing “marrow” in “bone derived macrophages” in the title. Please add the meaning of N/A. See table 3 as an example.

Table 2 – Once again the title in not descriptive of the table content. Also the APC number is not visible.

Overall the authors major exploit is the demonstration that miltefosine can be optimized (this concept itself is not novel and has been exploited for leishmania and other pathogens), they succeed in producing more active compounds that although active in vivo are not active upon oral administration a significant issue considering that Leishmania target product profile for a new drug favors orally available drugs. The lack of PK information on the selected compounds limits the comprehension of the potential of the compound and also prevents better planning for animal testing.

Author Response

We would like to thank the reviewers for the work they have completed in reviewing our manuscript of on “The Structure and Antiparasitic Activity Relationship of Alkylphosphocholine Analogues against Leishmania donovani’ .  We have addressed the individual points you have raised below. 

  1. The manuscript Structure and Antiparasitic Activity Relationship of Alkylphosphocholine Analogues against Leishmania donovani describes the evaluation of the anti-leishmanial potential of Alkylphosphocholine analogues. The premise for the article is scientifically sound, the search for Leishmania-optimized miltefosine derivatives.

Overall the message is clear and the manuscript is easy to read.

We are happy that the review understands the justification provided for our study recognised the message of our research to the readers of the journal, microorganisms

  1. The introductory section is objective and provides sufficient information for the comprehension of the scientific rational. Still some background justification for some options in the manuscript would be beneficial.

We have added the following sentences to lines 37-44 to give other options for identifying new antileishmanial drugs. ‘Ideally new drugs to treat leishmaniasis are required but this approach is limited by the fact that leishmaniasis is a neglected tropical disease, and will not make a good investment return for pharmaceutical companies [5]. Different approaches can be used to identify new drugs, for example,  repurposing clinically approved drugs for a different condition, identifying new druggable targets by having a better understanding the molecular targets of existing antileishmanial drugs, or by designing a better therapeutic based on structure activity studies using existing antileishmanial drugs.’

  1. Why to use antimony resistant parasites in this context?

Antimony resistance are widespread in VL endemic regions particularly in the Indian Subcontinent which are caused mainly by Leishmania donovani (Shaw et al., 2012). Therefore, we included data on the antileishmanial efficacy of our novel compounds against these type of strains as part of our screening studies.

What is the expected take home message from these specific parasites in the context of miltefosine derivatives?

This experiment was designed to validate APC12 as a lead compound and an alternative for Milt, and to show that novel compounds could be effective against strains with different inherent Sb-resistance as these types of parasites predominant in endemic regions, such as the Indian sub-continent.  We have tried to make this message clearer by changing the text in lines 207-29 with sentence starting from ‘Next, the antiparasitic activity of APC12, APC14 and APC16 against three Nepalese L. donovani clinical isolates with different inherent susceptibilities to antimony, reflective of strains present in endemic communities in the ISC was tested’.

  1. Would not be preferable to use miltefosine resistant parasites?

We agree with the reviewer that this data is appropriate for this study. We have now included this data using parasites with induced resistance to Milt, using parasites produced in our laboratory (Shaw et al., 2012, Antimicrob Agents Chemother 2019, 64, e00904-19).  Details of these strains have been added to the methods section (line 83-85), the results section (lines 231-239)  and the data is provided in Table 3.  

  1. Considering that the molecules are miltefosine derivatives and the changes on APCs subtle, what were the expectations on the possibility of synergism?

SAR studies in the literature highlight that subtle changes in the physical structure of APCs have significant  species-specific biological effect.  We have recently demonstrated this for the parasitic protozoan, Acanthamoeba, for which APC16 and not APC12 was active (Mooney et al., 2020) and have now applied the same to Leishmania.  Mechanistically, there are multiple reasons for this finding.  Interactions of APCs (with altered in the head, linker and tail) with cell membrane (with different phospholipids composition and numbers) have significant and unpredictable biological effects.  This is partly because death from APC are caused by the individual molecule, or as  grouped structures called micelles formed at concentrations called the critical micellar concentration (CMCs).  Again micelles formed from mixed APCs are produced at distinctly different CMC and have physical properties from their individual counterpart.  Their biological properties are a composite of these factors and the ratio of the two APCs.   Indeed, our study highlighted this; a strong link with the biological activities of physical properties such as alkyl carbon, a strong determinant of hydrophobicity and ease to from micelles and the biological properties for monotherapy and for the combined therapy, the activity was unpredictable and limited to define drug ratios.   We have identified ratios for mixed APCs that will inform drug formulation for a combined drug treatment. 

  1. Concerning the material and methods used I have some doubts/concerns: In line 88 please clarify if the plates were incubated at 25ºC or 27ºC.

We apologise for this typographical error, the plates were incubated at 25°C not 27°C.  The sentence now reads ‘The plate was incubated for 72 h at 25°C.’ instead of ‘The plate was incubated for 72 h at 25°C, 27°C’

  1. Concerning the macrophage infections for the purpose of evaluating the activity of the APCs against intracellular amastigotes I have some doubts. No reference to passages or days of culture are made. How many days of culture have the promastigotes that were used for the in vitro assays?

Late stage Leishmania promastigotes cultures at stationary phase for at least 2 days were prepared to a density of 1×10- 1×10and added to macrophages in in vitro studies.  In the lab, promastigotes are maintained for up to 25 passages.  We have sufficient data in our laboratory to know if we have any problems with infectivity.  

How was the parasite maintenance done (starting inoculum and maintenance schedule)?

Promastigotes were maintained in complete minimum media, RPMI or HOMEM supplemented with 20% v/v foetal calf/bovine serum at 25°C. Cultures were seeded at 2 x 105 parasites/ml in 10 ml of culture medium and sub-passaged similarly weekly.  The information on parasite culture has been changed from ‘Leishmania donovani LV82 (MHOM/ET/67:LV82), the Nepalese clinical Leishmania donovani isolates were cultured in complete RPMI 1640 medium (medium supplemented with 20% (v/v) heat-inactivated foetal calf serum, 1% (v/v) penicillin/streptomycin, 100 µg/ml and 1% (v/v) L-glutamine) at 25°C.’ toLeishmania donovani promastigotes of LV82 (MHOM/ET/67:LV82), the Nepalese clinical Leishmania donovani isolates were cultured in complete minimum media, RPMI 1640 or  HOMEM (supplemented with 20% (v/v) heat-inactivated foetal calf serum, 1% (v/v) penicillin/streptomycin, 100 µg/ml and 1% (v/v) L-glutamine) at 25°C.  The transgenic line cultures were further supplemented with Hygromycin B. Parasites were passaged weekly at a cell density of 2 x 105 cell/ml in a 10 ml volume of the appropriate medium in a 25 cm3 tissue culture flask’. (lines 89-94).  

  1. In the context of the macrophage infection, there is in line 94 there a reference to 1x106-1x107 as the number of parasites added to each well, please clarify this 10 fold difference in inoculum. If strain specific inoculums for infection ratios are used please describe the rational for choice of ratio of infection.

The infectivity of the luciferase expressing parasites varied, so we used different parasite: host cell ratios for the strains to ensure we had a good infectivity in studies.  Our aim was to obtain a bioluminescent signal that was in a comparable range for control data for the different strains.  We have altered the text from ‘In macrophage studies the method described used in previous studies was followed [14]. Briefly bone marrow derived BALB/c macrophages (1–2×105) in 100 µl complete RPMI 1640 medium were added to the wells of a 96 well plate and left to adhere for 24h at 37°C, 5% CO₂/95% air.  The medium was removed and 100 µl of the appropriate L. donovani luciferase-expressing promastigotes line (1×10⁶–1×10⁷) was added to each well. The plate was incubated for a further 24h, the medium removed, and, 100 µl of the medium alone (control) or appropriate APC compound was then added (0.01 µg/ml - 0.197 µg/ml, n = 3/treatment).’ to ‘The method described in previous studies for macrophage studies was follows [15]. Briefly bone marrow derived BALB/c macrophages (1×105) in 100 µl complete RPMI 1640 medium were added to the wells of a 96 well plate and left to adhere for 24h at 37°C, 5% CO₂/95% air.  The medium was removed and 100 µl of the appropriate L. donovani luciferase-expressing promastigotes line (parasite:host cell ratio 1:10, 1:20 or 1:40) was added to each well, the ratio used reflected the infectivity of the strain and ensured that control data gave a bioluminescent signal within the same rang . The plate was incubated for a further 24h, the medium removed, and each well washed with PBS to eliminate non-internalized parasites which was monitored microscopically.  When the wells were devoid of external parasites, 100 µl of the medium alone (control) or appropriate APC compound was then added (0.01 µg/ml - 0.197 µg/ml, n = 3/treatment)’ (lines 105-114).  

  1. Considering that 1-2x105 macrophages were used inoculums an infection ratio of 100 or 50 parasites per cell seems an over kill. Please justify this option. It is difficult to compare different strains/species because the infectivity of the parasites and the capacity to modulate the macrophage biology can have an impact in the sensitivity to drugs. Thus it is important to know the rational for the macrophage infections. How inter-strain infection normalization was achieved? Another aspect that I would like to see clarified was the 24 hour infection. Why 24 hour infection? Would not be better a short (4 hour) infection, than wait for 24 hours before adding the drug to enable the uptake of all the parasites and enable the differentiation. From our experience this originates much more reproducible data. As the test is described, the continuous infection for 24 hours, media removal and then treatment, there is the risk having drugs that are killing parasites that are attached to the surface or parasites that are undergoing differentiation. This is not amastigote killing and might contribute to false discovery rate because the objective of the assay is to detect compounds that act on amastigotes and not compounds that kill promastigotes or interfere in the differentiation.

The method used is for drug inhibition studies is based on studies using traditional Giemsa staining methods we (e.g. Carter et al. 2001 AAC 45:3555) or other researchers have used (e.g. Ashutosh et al., 2005,  AAC 49:3776).  Using luciferase expressing promastigotes we found that we had better infection rates if we incubated cells with parasites for 24 rather than 4 hours and it was  important in our studies to have a good signal for control parasites. We agree that we cannot rule out the possibility that some promastigotes could have been phagocytosised at the end of the 24 hour infection period, however this is true for all drug studies. We did complete preliminary studies where examined cells at 24 hours post-infection, using the traditional 24 well plate assay, where wells contained circular coverslips and attached cells were stained with Giemsa to allow us to visualize parasites. In these cells we saw amastigotes-like parasites within cells rather than promastigote-shaped parasites within the macrophages, suggesting that conversion was complete prior to drug administration. This gave us the confidence that the amastigotes were being killed in our experiments.

  1. Please define which was the statistic approach used from the graphpad software package to determine the IC50.

We apologise for our error in not reporting the software used for IC50 estimations.  The Grafit Software was used and not Graphpad. This error has been corrected in the revised manuscript.  The text that read ‘Percent viability values were used to calculate the IC50 using Graphpad® software (version 5.0)’ now readsPercent viability values were used to calculate the IC50 using Grafit® software (version 5.0)’ (lines 137-138).  

  1. For clarity would recommend a more appropriate description of the selectivity index (Line 120). Would suggest: “The selectivity indices (SI) were determined as the quotient between the cellular toxicity determined in the uninfected cells (CC50) and the anti-parasite activity in infected macrophages determined by the IC50.” Please confirm that no mistake exists in table 1. For example for APC12 if I use the data on the table to calculate SI I get 3870 (100,64/0,026) and not the 3535 reported.

This has been done. We apologise for the error in the SI values given.  The suggestion by the reviewer has been adopted but we have amended the text in lines 139-153 to:  

Selectivity indices (SI) were determined as: 

instead of    ‘The selectivity (SI50) indices were determined as the quotient of the IC50 of uninfected macrophages compared to infected macrophages; and the resistance (RI50) was determined as the quotient of the IC50s of drug-resistant strain (Sb-I or Sb-R) to the drug-sensitive strain (Sb-S) respectively.’   

  1. I had some difficulty to understand the rational for the in vivo experiments.

The rational is as follows (a) in vitro studies with promastigotes and amastigotes suggested that APC12 alone was a lead compound, (b) selectivity index suggested that it was safe for use in mice for in vivo animal experiments (c) the PK of APC12 whilst not determined in this study, was inferred to be superiors to APC, as [9.  Dorlo, T.P.; van Thiel, P.P.; Huitema, A.D.; Keizer, R.J.; de Vries, H.J.; Beijnen, J.H.; de Vries, P.J. Pharmacokinetics of miltefosine in Old World cutaneous leishmaniasis patients. Antimicrobial agents and chemotherapy 2008, 52,2855-2860] had suggested that molecules with short alkyl carbon bind less to serum proteins and have a short half life.

  1. Foremost I think that it was missing a PK evaluation to adjust the treatment scheme for the drug. At the moment in our team we are not testing in vivo any compounds without PK information. This helps to adjust the treatment scheme maximizing the chance of having good data. If the goal of IV administration was to confirm in vivo anti-parasitic capacity, why wait 7 days? This will probably “dilute” the effect by the multiplication of the surviving parasites, no? What is the rational for the 7 day waiting period, I would kill the animals maximum 4 days after administration. Please describe briefly the determination of parasitaemia. No reference is made in the section of material and methods of how critical micellar concentration was determined.

The method we used in in vivo studies is one we have used for a number of years to assess the antileishmanial activity of formulations against L. donovani, so we have a large amount of data on the expected parasite levels for controls and the antiparasitic activity of other antileishmanial drugs e.g. sodium stibogluconate (Carter et al.,  1988, J Pharm Pharmacol 40:370), amphotericin B (Mullen et al., 1988, AAC 42: 2722), miltefosine (Carter et al., 2003, AAC 47:1529) and paromomycin (Williams et al. 1988, J Pharm Pharmacol 50:1351).  The aim of the in vivo studies was to determine if APC12 was more active than APC16 and if the compound showed any site-specific effect.  We believe that an in vivo efficacy experiment is the most relevant to determine this.  We agree that pharmacokinetic studies are useful but (i) they depend on having an assay that is sensitive enough to detect levels in the site of interest and (ii) using more than one drug dose and/or more than one time point to assess drug levels.  In our opinion it used a lower number of animals to assess the potential of APC12 as an antileishmanial agent.  The CMC were obtained from the Anatrace product description sheet for respective APCs and we apologise for not being clear. We have now added the sentence ‘CMCs values were obtained from the Anatrace product description sheet for respective APCs.’  To the manuscript (lines 74-75) .

  1. Concerning the result section: Line 136-137, for accuracy it should read “Three the APC series with physical modifications at the head, tail and linker (Fig. 1) were screened for their antiparasitic activity against L. donovani (MHOM/ET/67:LV82) expressing luciferase”. Correct?

We apologise for this error.  The sentences has been updated with the number of analogues used in this study and it now reads ‘Nine APC series with physical modifications at the head, tail and linker (Fig. 1) were screened for their antiparasitic activity against L. donovani (MHOM/ET/67:LV82).’ instead of ‘Three APC series with physical modifications at the head, tail and linker (Fig. 1) were screened for their antiparasitic activity against L. donovani (MHOM/ET/67:LV82)’  at lines 173-174.

  1. To my understanding all the tests were performed using the luciferase transfected parasites. It is different from doing in Wild Type because the transfection process will involve a selection that might alter the basic characteristics of the parental population.

We confirm that all the parasites used were transgenic luciferase expressing parasites.  While it is possible that this process can alter the basic biology of the transgenic parasites, relative to the parental population, preliminary cytotoxicity and viability assessments with the alamar blue produced comparable results for the ‘transgenic‘ and ‘wild type’ parasites (data not shown).  We concluded that the process had no effect on the cell line used in this study.  

  1. Line 143 it states that “APC14 and APC12, increased the biological activity of the resulting compound significantly” How was this significance calculated, what was the statistical test?

This was suggested by another reviewer and we have provided a separated section to make clear the test used to test for significance in this study.  A new section, titled ‘Section 2.4. Data analysis’ has been added to the Methods section in the resubmitted manuscript (Lines 165-169). 

  1. Table 1, 2 and 3 all have statistical analysis but I could not find reference to any statistical test in methods section.

This is now been detailed in section 2.4 (lines 287-291).

  1. Along the test significance claims are made without the appropriate support for what is being compared. For example, Line 150 “…APC12 tail (APC11UPC; Fig 1), allowed the formation of lamellar structures and not micelles [18] significantly reduced the efficacy related to APC12 (Table 1 and Fig S1)”. When I look to Table 1 the statistical analysis present is related to APC16, so not only I have no way of knowing what is being compared, (Promastigotes? Amastigotes?) looking at the amastigote data the most relevant I would say they are very similar. This lifts another question, if I consider Selectivity index APC11UPC is the best derivative, why was not considered for further studies? As stated above I see no added value in the testing to Sb resistant strains, more so when the studies are performed in transfected parasites.

We apologise for the lack of clarity.  We have now amended the manuscript to make clear what test was used and the variables being tested.  We agree with the reviewer that APC11UPC was the best derivative.  Our choice of compound for detailed investigation was in part determined by cost and availability and we deemed that its use as a choice molecule was not practical. 

We have also added data for the antileishmanial efficacy of APC analogues, APC12, APC 14 and APC16 against Milt resistant parasites, produced against the strains with different inherent susceptibilities to Sb. See Table 3.

  1. Concerning section 3.2 please try to explain in the discussion section the interesting difference between DA and DAB in amastigotes.

The activity of DAB and DA has been discussed in the discussion section.  See Line 396-404.

  1. The synergism section is interesting showing that the synergism correlates seen for promastigotes are not always reportable for the intracellular assay suggesting as the authors state that combination studies should focus on the intracellular stage (line 268). If the N/A I table 1 means “not active” than APC11UPC would have been a very nice compound to evaluate in the context of synergism because is active but not forming micelles.

We apologise for not defining NA and we have now defined NA in the foot note of relevant tables.  NA represent CMCs values not available in the MSDS sheet provided by Anatrace.  The use of APC11UPC in combination assay is straight forward, but due to prevailing restrictions to lab access in response to the COVID 19, with our labs in lock-down, these experiments cannot be done at this stage.  This investigation will form the basis of our ongoing studies on this molecule.  We thank the reviewer for this observation.

  1. The in vivo data could be improved if PK of the APC12 was available, the fact that APC12 did not work upon oral administration is disappointing and limits the value of APC12 as oral drugs are a priority in Leishmania drug development. The use of a single administration is baffling, I have used miltefosine 20 mg/kg for 20 days without any side effect and has complete clearance of infection in all quantifiable compartments. I would recommend quantify the compound of interest in the blood after administration, perform a PK and then adjust the treatment accordingly, maybe bi-daily administration is what is required to have sufficient bio-accumulation in the blood and probably one administration is not sufficient for this to happen. A PK would greatly improve the understanding of the apparent lack of activity upon oral administration. The fact that works after IV administration is a confirmation that if the molecule is on the blood it would work so this should be taken as encouraging.

We agree that pharmacokinetic studies are useful but (i) they depend on having an assay that is sensitive enough to detect levels in the site of interest and (ii) using more than one drug dose and/or more than one time point to assess drug levels.  This is something that we will consider in the future.

  1. Supplemental information: It is very nice to see all the curves. In figure S2 the curve for APC12 in graph D is not sigmoidal having a plateau at 50%. How was the EC50 determined in this situation?

We apologise for this error.  We have now added the correct file to the supplementary data.  This is shown in Figure S2 in the resubmitted supplementary materials.

  1. Minor –All the EC50 and EC90 in the text and tables should be normalized to the same format using either two or three decimal cases. Sometimes you have standard deviation with more decimal cases that the actual reported value. For example line 140 “6.3 ± 0.10 µg/ml” or “0.90 0.10 µg/ml”.

We apologise for this formatting error.  We have changed the way we cite the EC values so they are in the same format.

  1. Table 1 – The title is not completely descriptive of the table, much more than just SI and EC50 are depicted. It is also missing “marrow” in “bone derived macrophages” in the title. Please add the meaning of N/A. See table 3 as an example.

A more comprehensive descriptive title has been provided for Table  1.  We have changed XXXX to ‘The activity of APCs against L. donovani parasites with different inherent susceptibilities to antimony i.e. Sb sensitive, Sb intermediate or Sb resistant, at the promastigotes and intracellular amastigotes stage’ (line 217-222)

  1. Table 2 – Once again the title in not descriptive of the table content. Also the APC number is not visible.

A more comprehensive descriptive title has been provided for Table 2.  We have changed XXXX to ‘The activity of L. donovani parasites with different inherent Sb susceptibilities which had induced resistance to Milt i.e. Milt Sb sensitive, Milt Sb intermediate or Milt Sb resistant, at the promastigotes and intracellular amastigotes stage’ (lines 416-417)

  1. Overall the authors major exploit is the demonstration that miltefosine can be optimized (this concept itself is not novel and has been exploited for leishmania and other pathogens), they succeed in producing more active compounds that although active in vivo are not active upon oral administration a significant issue considering that Leishmania target product profile for a new drug favors orally available drugs. The lack of PK information on the selected compounds limits the comprehension of the potential of the compound and also prevents better planning for animal testing.

Research on the compound to increase bioavailability, which included developing an assay for estimating drug levels in serum for PK analysis are ongoing.

Reviewer 3 Report

The paper by Ahmed H. et al. describes the activity of several miltefosine derivatives against in vitro and in vivo Leishmania infections. These derivatives differ in the length and structure of the alkyl carbon chains and also in the characteristics of the polar heads of the molecules. APC12 analog is 4-10 fold more active than miltefosine against in vitro cultured intracellular amastigotes (depending on the parasite strain used) and its selectivity index is about 60 fold better than the parental compound. However, APC12 does not cause any significant reduction in the number of parasites in the bone marrow of infected mice when given by the intravenous route and is less effective than miltefosine by the oral route.

Major concern.

Even though these compounds show very good activity against in vitro cultures of the different forms of Leishmania parasites, their activity in mice does not support any improvement over the parental molecule miltefosine.

Minor concerns.

Line 77. “integration into the ribosomal DNA locus” Even though the fragments integrate into the chromosomes (DNA) the locus is named “ribosomal RNA locus”.

Lines 94-96. After incubation of the cells with promastigotes, the medium is removed and then a new medium is added either with the compound to be tested or with medium alone. Usually, one or several washes are needed to effectively eliminate any non-internalized parasites and then the plate must be checked (by microscopy) to confirm the absence of these external parasites. Neither of these activities is described in the protocol. The authors should clarify this procedure.

Line 138. “mean IC50 ± SDS” I understand that SDS means standard deviation but using the term SDS is confusing. SD is more correct.

Line 179. “Resistance indexes (RI50) for promastigotes and amastigotes ranged from 0.48-1.77 and 0.77-2.33 respectively, showed that the RI50 value reflected antimony resistance of the strain” The concept of resistance index is not defined in the manuscript. If this term refers to the resistance of the strains against antimony the sentence seems to have no meaning.

Lines 209 and 210. “Suggesting that identification of synergistic combination ratios was not a function of their physical properties, unpredictable and a systematic approach was required for identification” This sentence needs rewriting.

Line 237. “milt-analogue, APC16” Even though it is not clearly stated in the manuscript, according to the structure shown, APC is miltefosine, not a milt-analogue.

Line 255. “with 21, 18, 16, 18 alkyl carbon atoms tails” 18 is repeated and does not describe the number of atoms in perifosine.

Author Response

We would like to thank the reviewers for the work they have completed in reviewing our manuscript of on “The Structure and Antiparasitic Activity Relationship of Alkylphosphocholine Analogues against Leishmania donovani’ .  We have addressed the individual points you have raised below.

The paper by Ahmed H. et al. describes the activity of several miltefosine derivatives against in vitro and in vivo Leishmania infections. These derivatives differ in the length and structure of the alkyl carbon chains and also in the characteristics of the polar heads of the molecules. APC12 analog is 4-10 fold more active than miltefosine against in vitro cultured intracellular amastigotes (depending on the parasite strain used) and its selectivity index is about 60 fold better than the parental compound. However, APC12 does not cause any significant reduction in the number of parasites in the bone marrow of infected mice when given by the intravenous route and is less effective than miltefosine by the oral route.

Major concern.

  1. Even though these compounds show very good activity against in vitro cultures of the different forms of Leishmania parasites, their activity in mice does not support any improvement over the parental molecule miltefosine.

We share similar concern as the reviewer.  Our data indicated reduced bioavailability.  Studies to improve bioavailability, estimate PK, and formulate the compound within liposomes to improve activity and development of better in vitro systems to screen compounds.  We think that our studies highlight this, but also show how SAR can be used to identify active compounds.

Minor concerns. 

  1. Line 77. “integration into the ribosomal DNA locus” Even though the fragments integrate into the chromosomes (DNA) the locus is named “ribosomal RNA locus”.

We apologise for this error.  The sentence now reads ‘integration into the ribosomal RNA locus’ instead of ‘integration into the ribosomal DNA locus’. Line 88 in the resubmitted manuscript.

  1. Lines 94-96. After incubation of the cells with promastigotes, the medium is removed and then a new medium is added either with the compound to be tested or with medium alone. Usually, one or several washes are needed to effectively eliminate any non-internalized parasites and then the plate must be checked (by microscopy) to confirm the absence of these external parasites. Neither of these activities is described in the protocol. The authors should clarify this procedure.

We apologise that the brief summary has excluded some vital and important information of our experimental protocols.  We have now identified these deficiencies and made the experimental protocol more detailed in the resubmitted manuscript.    The sentence now reads ‘The plate was incubated for a further 24h, the medium removed, and each well washed with PBS to eliminate non-internalized parasites which was monitored microscopically.  When the wells were devoid of external parasites, 100 µl of the medium alone (control) or appropriate APC compound was then added (0.01 µg/ml - 0.197 µg/ml, n = 3/treatment).’ instead of ‘The plate was incubated for a further 24h, the medium removed, and 100 µl of the medium alone (control) or appropriate APC compound was then added (0.01 µg/ml - 0.197 µg/ml, n = 3/treatment).’. See Lines 127-134 in the submitted manuscript.

  1. Line 138. “mean IC50 ± SDS” I understand that SDS means standard deviation but using the term SDS is confusing. SD is more correct.

The error has been corrected and the sentence  now reads ‘mean IC50 ± SD’ instead of ‘mean IC50 ± SDS’.  Line 175 in the submitted manuscript

  1. Line 179. “Resistance indexes (RI50) for promastigotes and amastigotes ranged from 0.48-1.77 and 0.77-2.33 respectively, showed that the RI50 value reflected antimony resistance of the strain” The concept of resistance index is not defined in the manuscript. If this term refers to the resistance of the strains against antimony the sentence seems to have no meaning.

Resistance index is now defined clearly defined in Lines 152 (submitted manuscript).  We also accept the point made by the reviewer that the use of resistance index was inappropriate, as the resistant parasites were not resistant to APCs but to Sb.  As this reflects cross resistance, we have added a new terminology and definition for clarity.  The revised sentence now reads ‘and the resistance (RI50) and cross resistance (CRI) indexes were determined as the quotient of the IC50s of drug-resistant strain (Sb-I or Sb-R) to the drug-sensitive strain (Sb-S) to Sb and another substance, e.g. APC, not involved in the induction of resistance respectively’ instead of ‘and the resistance (RI50) was determined as the quotient of the IC50s of drug-resistant strain (Sb-I or Sb-R) to the drug-sensitive strain (Sb-S) respectively.’.

We have amended the text in lines 164-167 to:

Cross resistance (CRI50) indexes were determined as:

We have also amended the sentence which now reads ‘Cross-resistance indexes (CRI50) for Sb-resistant promastigotes and amastigotes to APCs ranged from 0.48-1.77 and 0.77-2.33 respectively, showing that the CRI50 value reflected antimony resistance of the strain (Table 2; Fig. S2)’ instead of ‘Resistance indexes (CRI50) for promastigotes and amastigotes ranged from 0.48-1.77 and 0.77-2.33 respectively, showing that the RI50 value reflected antimony resistance of the strain (Table 2; Fig. S2)’. See lines 232-234

  1. Lines 209 and 210. “Suggesting that identification of synergistic combination ratios was not a function of their physical properties, unpredictable and a systematic approach was required for identification” This sentence needs rewriting.

This sentence have been rewritten and now reads ‘suggesting that identifying synergistic combinations, was unpredictable, and unrelated to the physical properties of APCs, but required systematic screening of all possible drug combination ratios within the physiological level for the model organism.’ instead of ‘suggesting that identification of synergistic combination ratios was not a function of their physical properties, unpredictable and a systematic approach was required for identification’  See lines 284-286

  1. Line 237. “milt-analogue, APC16” Even though it is not clearly stated in the manuscript, according to the structure shown, APC is miltefosine, not a milt-analogue.

We agree with the reviewer. APC16 is miltefosine and while this was clear elsewhere in the manuscript, consistency was not maintained in this section and wrongly presented it as an analogue.  The sentence has been modified and now reads ‘Milt (APC16)’ instead of ‘milt-analogue, APC16’.  See line 314.

  1. Line 255. “with 21, 18, 16, 18 alkyl carbon atoms tails” 18 is repeated and does not describe the number of atoms in perifosine.

We note that perifosine has 25 carbon atoms.  The sentence presented refers to the alky carbon chain of the tail comprised of 18 carbon atoms. 

Round 2

Reviewer 1 Report

The description of significance in the tables is still inappropriate,

in line 160 choose only

on the table description simply say "Significantly different"

in the table make sure to have the same number of decimals for all entries and to report standard errors different from "0.00"

Author Response

We would like to thank the reviewers for the work they have completed in reviewing our manuscript of on “The Structure and Antiparasitic Activity Relationship of Alkylphosphocholine Analogues against Leishmania donovani’ . We have addressed the individual points you have raised below.

  1. The description of significance in the tables is still inappropriate, in line 160 choose only

on the table description simply say "Significantly different" 

 This has been changed. 

  1. in the table make sure to have the same number of decimals for all entries and to report standard errors different from "0.00"

We are not sure what the reviewer would like us to do.  We have given SE to two significant figures.  Therefore, SE that was in the range 0.000-0.004 is reported as 0.00.  If a SE was in the range 0.005-0.009 then we reported the error as 0.01.  We could change all the values to three significant figures but this would be beyond the sensitivity of the assays and make the tables too cumbersome.

Reviewer 2 Report

Overall the revised manuscript is a clarified version of the original submission with new data added (the interesting data on miltefosine resistant parasites) that improves the manuscript. Still, my main concern remains. The absence of PK limits the value of the in vivo data. At the moment it is not known if the compound does not work in vivo (oral administration) or just a matter of bio-availability. I firmly disagree with the authors on this aspect. It is my conviction that in the 21st century with the tools available, the determination of bio-availability of a drug should preclude the activity testing. This can be done with 6 BALB/c mice 20 mg/kg a single oral administration, and recovery of blood from the tail in staggered animals at specific time points. This will give valuable information that will enable to proper plan the in vivo experiment. Is not a full PK/PD evaluation, it is just a quick PK (SNAP-PK) that will inform on how much compound is in the blood and enable to quickly evaluate if a compound has the potential to work in vivo. If a compound has no/reduced oral availability why to infect animals and test it? The main limitation could be the lack of capacity to quantify the compound in the blood. But this can be overcome. Still, in the absence of PK the animals were infected, treated and the information generated was mostly inconclusive. At the moment we do not know if the regimen of treatment was appropriate, or if the drug is orally available at all. And no proof of in vivo superiority when compared to miltefosine exists. In my opinion this would be a difference maker for the overall interest in the manuscript providing a possible new orally available lead compound.

Other minor concerns remain, the use of different amounts of parasites to ensure good infectivity for the in vitro activity experiments. This introduces variability to the experiments limiting strain to strain comparisons, that are always difficult to perform. The experimental setup chosen (24 hours infection followed by treatment) . The 24 hours infection followed by immediate treatment, even if amastigote like parasites are seen in the macrophages, many will be still in the differentiation process, this might originate a false activity spectrum that is associated with a drug that interferes with the differentiation process. In vivo this will not be relevant because promastigote infections will only happen in the initial inoculum. A four-hour infection followed by a waiting period of 24 hours before treatment would give higher confidence concerning amastigote specific activity.

Author Response

We would like to thank the reviewers for the work they have completed in reviewing our manuscript of on “The Structure and Antiparasitic Activity Relationship of Alkylphosphocholine Analogues against Leishmania donovani’ . We have addressed the individual points you have raised below.

  1. Overall the revised manuscript is a clarified version of the original submission with new data added (the interesting data on miltefosine resistant parasites) that improves the manuscript.

We are pleased that we have addressed a lot of your comments.   Thank you for your input as this has improved the quality of our manuscript.

  1. Still, my main concern remains. The absence of PK limits the value of the in vivo data. At the moment it is not known if the compound does not work in vivo (oral administration) or just a matter of bio-availability. I firmly disagree with the authors on this aspect. It is my conviction that in the 21stcentury with the tools available, the determination of bio-availability of a drug should preclude the activity testing. This can be done with 6 BALB/c mice 20 mg/kg a single oral administration, and recovery of blood from the tail in staggered animals at specific time points. This will give valuable information that will enable to proper plan the in vivo experiment. Is not a full PK/PD evaluation, it is just a quick PK (SNAP-PK) that will inform on how much compound is in the blood and enable to quickly evaluate if a compound has the potential to work in vivo. If a compound has no/reduced oral availability why to infect animals and test it? The main limitation could be the lack of capacity to quantify the compound in the blood. But this can be overcome. Still, in the absence of PK the animals were infected, treated and the information generated was mostly inconclusive. At the moment we do not know if the regimen of treatment was appropriate, or if the drug is orally available at all. And no proof of in vivo superiority when compared to miltefosine exists.  In my opinion this would be a difference maker for the overall interest in the manuscript providing a possible new orally available lead compound.

We understand the comments made by the reviewer and know that PK data would strengthen the manuscript.  At present because of the corona virus pandemic we are unable to get into the laboratory and have no indication of when this would be possible.  We recognize the lack of PK data has on our ability to make firm conclusions on the why the novel compound APC12 is not as effective as miltefosine by the oral route. We have modified the discussion (lines 492-493) to highlight these limitations and ensure that the conclusion made is reflective on the data presented.  

  1. Other minor concerns remain, The use of different amounts of parasites to ensure good infectivity for the in vitro activity experiments.   This introduces variability to the experiments limiting strain to strain comparisons, that are always difficult to perform.  The experimental setup chosen (24 hours infection followed by treatment). The 24 hours infection followed by immediate treatment, even if amastigote like parasites are seen in the macrophages, many will be still in the differentiation process, this might originate a false activity spectrum that is associated with a drug that interferes with the differentiation process.  In vivo this will not be relevant because promastigote infections will only happen in the initial inoculum. A four-hour infection followed by a waiting period of 24 hours before treatment would give higher confidence concerning amastigote specific activity.

We agree with the reviewer that differences in infectivity between strains are always a problem and  a limitation in in vitro studies.  We have used an experimental method that has been used in our laboratory for a while and published in previous drug screening studies.  I am not sure what we can do to address this issue as our laboratories are closed because of the corona pandemic and we do not know when they will be open again.  Therefore, we cannot carry out an in vitro study to determine if the suggested protocol has any impact on drug efficacy.  

Round 3

Reviewer 2 Report

I fully understand the limitations of access to the laboratory. Thus, I have no further remarks to improve the manuscript in its current form. Just a final minor note, the changes in the discussion are in line 435-436 and not 492-493 (This is a new reference added).